# Economic Trends in Commonly Used Drugs for Spinal Fusion and Brain Tumor Resection: An Analysis of the Medicare Part D Database

**DOI:** 10.3390/biomedicines11082185

**Published:** 2023-08-03

**Authors:** Jagroop Doad, Nithin Gupta, Lydia Leavitt, Alexandra Hart, Andrew Nguyen, Shawn Kaura, Frank DeStefano, Edwin McCray, Brandon Lucke-Wold

**Affiliations:** 1Leon Levine Hall of Medical Sciences, School of Osteopathic Medicine, Campbell University, 4350 US Hwy 421 S, Lillington, NC 27546, USA; 2College of Medicine, University of Illinois, 1601 Parkview Ave., Rockford, IL 61107, USA; 3Lake Erie College of Osteopathic Medicine at Seton Hill, Lynch Hall, 20 Seton Hill Dr, Greensburg, PA 15601, USA; 4College of Medicine, University of Florida, 1600 SW Archer Rd., Gainesville, FL 32610, USA; 5Department of Neurological Surgery, University of Kansas Medical Center, 2060 W 39th Ave., Kansas City, KS 66160, USA; 6Department of Orthopedic Surgery, College of Medicine, University of Arizona, 1501 N Campbell Ave., Tucson, AZ 85724, USA; 7Department of Neurosurgery, University of Florida, 1600 SW Archer Rd., Gainesville, FL 32610, USA

**Keywords:** neurosurgery, spinal fusion, brain tumor resection, Medicare, economics, pharmacology, health system, surgical system

## Abstract

With the incidence of central and peripheral nervous system disorders on the rise, neurosurgical procedures paired with the careful administration of select medications have become necessary to optimize patient outcomes. Despite efforts to decrease the over-prescription of common addictive drugs, such as opioids, prescription costs continue to rise. This study analyzed temporal trends in medication use and cost for spinal fusion and brain tumor resection procedures. The Medicare Part B Database was queried from 2016 to 2020 for data regarding spinal fusion and brain tumor resection procedures, while the Part D Database was used to extract data for two commonly prescribed medications for each procedure. Pearson’s correlation coefficient and linear regression were completed for the analyzed variables. The results showed a significant negative correlation between the number of spinal procedure beneficiaries and the cost of methocarbamol, as well as between the annual percent change in spinal beneficiaries and the annual percent change in oxycodone cost. Linear regression revealed that oxycodone cost was the only parameter with a statistically significant model. Moving forward, it is imperative to combat rising drug costs, regardless of trends seen in their usage. Further studies should focus on the utilization of primary data in a multi-center study.

## 1. Introduction

With 20% of the United States population expected to be above the age of 65 by 2030, there will be an increasing burden placed on the US healthcare system [1]. Currently, the age group of 65 or older represents 15% of the population but already accounts for almost half of the noted healthcare expenditures per year [2]. This dramatic rise in the aging population inevitably yields an increasing incidence of disease, including brain and central nervous system (CNS) pathologies as well as lumbar spine fusion procedures for degenerative disc disease [3,4].

Two of the most common procedures performed by neurosurgeons—spinal fusion and tumor resection surgeries—have also exhibited a large increase in the number of operations performed over the past decade [5,6]. For example, the volume of elective lumbar fusion procedures increased by 62.3% from 2004 to 2015, with increases being greatest in the elderly population of 65 and older [6]. Similarly, the incidence of brain tumors, as analyzed by ICD-10 codes C70.0–C72.9 (C70, malignant neoplasm of the meninges; C71, malignant neoplasm of the brain; C72, malignant of the spinal cord, cranial nerves, and other parts of the CNS), increased by 94.3% from 1990 to 2019 [5]. As such, there have been advances in pre-operative planning, minimally invasive techniques, and cutting-edge technologies that have drastically improved neurosurgical patient outcomes. Although these efforts to improve patient outcomes have been successful, the cost of these procedures has risen. A study by Martin et al. noted that aggregate costs of lumbar fusion procedures were shown to rise by 177% from 2004 to 2015 [6]. Similar trends were observed in brain tumor resection procedures over the previous two decades [7]. Thus, it becomes clear that, although the rising burden of neurosurgical disease in the United States is being met with exceptional patient outcomes, the cost has increased.

This rise in neurosurgical procedure costs can partially be attributed to variabilities in drug prices along with other noted factors such as length of stay, type of procedure, and geographic location factors. A recent study assessed the likelihood that a certain neurosurgical procedure would occur at different neighboring hospitals and termed that likelihood the “competitiveness” of the hospital. Historically, more complex procedures have been shunted to large, tertiary care medical centers by way of greater access to resources. This study also noted that higher charges of admission for in-hospital cranial procedures were significantly correlated with this competitiveness [8]. This could potentially point to a significant concept suggesting that costs for medical procedures increase in more affluent areas. Furthermore, as the number of procedures increases, as is to be expected with an aging population, the cost of drugs and procedures will rise due to the increased demand. This trend can be observed in commonly used drugs such as Alteplase, a powerful thrombolytic agent and a biosynthetic form of human tissue-type plasminogen activator (t-PA) with thrombolytic function. The Centers for Medicare and Medicaid Services (CMS) announced that Alteplase cost $30.50 in 2005, but the same drug cost $64.30 in 2014 [9]. This increase of 111% in cost was noted to be higher than the increase seen globally for drug prices during this time period, which was noted to be about 30%. This finding was shown to be significant due to the findings reported by Otite et al., who noted that IV tPA and mechanical thrombectomy utilization increased in acute ischemic stroke cases over the last decade [10].

Two of the most frequently utilized drugs in spinal fusion procedures are oxycodone, a semi-synthetic opioid, and methocarbamol. Oxycodone, a powerful analgesic, is used both peri-operatively and post-operatively as a powerful pain reliever; however, it has a strong addictive correlation associated with its usage. Methocarbamol is a skeletal muscle relaxant that functions to minimize muscle spasms and is commonly prescribed post-operatively in spinal fusion operations. For brain tumor resections, dexamethasone and levetiracetam are commonly used therapies, as they have been shown to play a role in directly inhibiting tumor proliferation and increasing sensitivity to chemotherapy in vitro in retrospective studies [11]. Levetiracetam, specifically, is also one of the most commonly prescribed prophylactic anti-epileptic drugs in brain tumor resections among neurosurgeons [12], possibly due to its low side effect profile, unique mechanism of action, and low rate of drug interactions.

As the number of neurosurgical procedures continues to grow each year, it is imperative to combat the rising costs of drugs to decrease barriers to patient care. Thus, this study seeks to elucidate the temporal trends in spinal fusion procedures and brain tumor resections and to also explore their relationships to the cost of drugs. In doing so, we hope to provide insight that will guide efforts to decrease the cost of neurosurgical operations.

## 2. Materials and Methods

### 2.1. Data Source

Data were obtained from the Medicare Prescriber Public Use Files: Part D from the Centers for Medicare and Medicaid Services (CMS) from 2016 to 2020 [13]. This database consists of information on prescription drugs prescribed to beneficiaries enrolled in Medicare Part D by physicians, sorted by provider and drug. Additionally, data were obtained from the Medicare Prescriber Public Use Files: Part B from 2016 to 2020 [14]. This database consists of information on the services and procedures provided to Medicare Part B beneficiaries by physicians, sorted by provider and procedure. Publicly available data was utilized; hence, no institutional review board approval was needed for this retrospective study.

### 2.2. Data Analysis

Two commonly prescribed medications for spinal fusion surgery, oxycodone and methocarbamol, and two commonly prescribed medications for brain tumor resection, dexamethasone and levetiracetam, were chosen for analysis. The Medicare Part D database was filtered by provider type for neurosurgeons and by generic drug name for oxycodone, methocarbamol, dexamethasone, and levetiracetam. The total number of prescriptions for the four drugs was calculated for each year. The Medicare Part B database was queried for various spine fusion and brain tumor resection procedures performed by neurosurgeons from 2016 to 2020.

### 2.3. Statistical Analysis

All statistics and calculations were performed using Microsoft Excel for Office 365. Pearson’s correlation coefficient was computed to assess the individual relationships of each variable—the total number of spinal procedure beneficiaries, the total number of spinal procedure services, and the average Medicare payment for spinal procedures—to the number of prescriptions and cost of oxycodone and methocarbamol. Furthermore, an additional Pearson’s correlation coefficient was computed to assess the relationship between the annual percent change of each of the variables analyzed and the annual percent change in the number of prescriptions and cost of oxycodone and methocarbamol. Identical methods were utilized for brain tumor resection procedures with dexamethasone and levetiracetam. A linear regression was performed to predict if the total number of beneficiaries, total number of services, and average Medicare payment were associated with the drug cost for each of the four prescriptions. The analysis of the included variables was calculated for all years together in order to show that significant relationships were truly predicted by the predictor variables. In addition to this, an analysis of variance was also conducted to verify that significant findings were indeed due to the generated model rather than some unexplained random error. Additionally, the standardized coefficients for the regression model were calculated to determine the significance of the predictor variables. Lastly, an effect size was calculated using the B (unstandardized) and beta (standardized) values in order to calculate the strength of our linear regression coefficients. All hypothesis testing was 2-tailed, with significance being set at *p* < 0.05. Statistical analysis including Pearson’s correlation and linear regression are included in the Appendix A.

## 3. Results

Using the Medicare Part D database, the annual total number of prescriptions and the average cost for oxycodone, methocarbamol, dexamethasone, and levetiracetam made by neurosurgeons were calculated from 2016 to 2020 (Table 1). The number of oxycodone prescriptions decreased by 36.7% from 107,663 to 68,171, and the cost of oxycodone decreased by 42.7% from $2567.31 to $1471.2. Conversely, methocarbamol prescriptions increased by 27.2% from 11,037 to 14,042, and the cost of methocarbamol increased by 27.7% from $301.47 to $385.27. Dexamethasone prescriptions decreased by 8.6% from 5292 to 4837; however, the cost of dexamethasone increased by 55.6% from $287.15 to $446.83. Finally, levetiracetam prescriptions decreased by 27.8% from 14,251 to 10,295, while the cost of levetiracetam increased by 9.2% from $1056.25 to $1154.25.

Using the Medicare Part B database, the annual total number of beneficiaries and average cost were calculated for both spinal fusion procedures and brain tumor resection procedures (Table 2). The total number of beneficiaries for spinal fusion procedures decreased by 17.7%, from 80,156 to 65,987, from 2016 to 2020. The average cost for spinal fusion procedures increased by 0.4%, from $4914.08 to $4937.06, over the time period. The total number of beneficiaries for brain tumor resection procedures decreased by 8.6%, from 2641 to 2416, from 2016 to 2020. The average cost for brain tumor resection procedures decreased by 5.5%, from $10,111.51 to $9558.81, over the time period.

The temporal trends for percent difference in the total number of beneficiaries, the average cost of the procedure, and the average cost of the two commonly used drugs for spinal fusion operations are illustrated in Figure 1. This data indicated that, although the cost of spinal procedures has remained relatively stable with a decrease in beneficiaries, the cost of oxycodone and methocarbamol prescribed by neurosurgeons has increased. The trends for percent difference in total number of beneficiaries, average cost of procedure, and average cost of the two commonly used drugs for brain tumor resection are illustrated in Figure 2. It shows that dexamethasone and levetiracetam followed similar trends but seemed to fluctuate recently in a reciprocal manner when compared to brain tumor resection beneficiaries and the cost of the operation.

A Pearson correlation was performed to assess the strength of the association between variables associated with spinal fusion procedures. The analysis demonstrated a statistically significant negative correlation between the total number of spinal procedure beneficiaries and the cost of methocarbamol, *r* = −0.89 (*p* = 0.042). However, there was no significant correlation between the total number of spinal procedure beneficiaries and the number of methocarbamol prescriptions. There was a weakly negative but non-significant correlation between the total number of spinal procedure services and the cost and number of prescriptions for methocarbamol. There was a weakly positive but non-significant correlation between the total number of spinal procedure beneficiaries and the number of spinal procedure services when compared with the cost or number of prescriptions for oxycodone.

When comparing annual percent changes of either services or beneficiaries with annual percent changes in the number of prescriptions and cost for oxycodone and methocarbamol, there was a significant negative correlation seen between the annual percent change in the total number of spinal procedure beneficiaries and the annual percent change in oxycodone cost, *r* = −0.97 (*p* = 0.032). There was also a significant negative correlation seen between the percent change in the annual total number of spinal procedure services and the annual percent change in oxycodone cost, *r* = −0.98 (*p* = 0.021). No other notably significant relationships were found.

A Pearson correlation was also performed to assess the strength of association between variables associated with brain tumor resection procedures. The analysis demonstrated a statistically significant negative correlation between the total number of brain tumor resection procedure beneficiaries and the cost of levetiracetam, *r* = −0.97 (*p* = 0.006). There was also a statistically significant negative correlation between the total number of brain tumor resection procedure services and the cost of levetiracetam, *r* = −0.98 (*p* = 0.003).

When comparing annual percent changes of either services or beneficiaries with annual percent changes in the number of prescriptions and cost of levetiracetam and dexamethasone, there was a significant negative correlation seen between the annual percent change in the total number of brain tumor resection procedure beneficiaries and the annual percent change in the cost of levetiracetam, *r* = −0.97 (*p* = 0.027). There was also a significant negative correlation seen between the annual percent change in the total number of brain tumor resection procedure services and the annual percent change in the cost of levetiracetam, *r* = −0.98 (*p* = 0.018). No other notably significant relationships were found.

Finally, multiple linear regression analysis was completed to determine if the total number of beneficiaries, total number of services, and average Medicare payment were associated with drug costs. Fitted regression models were created for methocarbamol, oxycodone, dexamethasone, and levetiracetam. Oxycodone (Table 3) was the only drug of the four that showed an overall statistically significant regression model (*p* = 0.032). It was found that the total number of beneficiaries and services was significantly associated with the cost of oxycodone (*p* = 0.019). However, the average Medicare payment was not associated with the cost of oxycodone. The standardized effect sizes for the number of beneficiaries, services, and payment were calculated to be 2.646, 2.468, and 0.088, respectively, suggesting a similar effect size for the cost of oxycodone.

## 4. Discussion

In this study, we assessed the temporal trends in commonly used drugs for both spinal fusion and brain tumor resection surgeries. By focusing on the most prescribed medications for spinal fusion surgery and brain tumor resection, we aimed to capture trends and characteristics relevant to a significant portion of patients undergoing these procedures. Although it is important to acknowledge that other medications could also be used in these procedures, introducing a broader range of medications could potentially introduce further complexities and reduce the specificity of our findings. Our analysis demonstrated that there was a statistically significant negative correlation between the total number of spinal procedure beneficiaries and the annual percent change in oxycodone and methocarbamol costs. Additionally, we found that there was a statistically significant correlation between the total number of brain tumor resection procedure beneficiaries and services when each was compared to the cost of levetiracetam. When linear regression analysis was applied, oxycodone was the only drug that was significantly associated with the number of beneficiaries and services.

Oxycodone is one of the most commonly prescribed drugs for neurosurgical patients. In Rautalin et al.’s 2021 study of in-hospital trends for neurosurgery patients, oxycodone comprised 49% of all opioids prescribed in the postoperative period [15]. Further, this study revealed that, although opioids did decrease across their 12-year study period, the majority of the decrease came from weaker-strength opioids. A large decrease of 82% was seen in the overall use of weak opioids, whereas the overall use of strong opioids decreased by only 12% during the 12 years. With strong opioids such as oxycodone still playing such an integral role in spinal procedures, it is important to consider alternatives with less potential for long-term side effects, such as addiction. Our study findings can be situated within the broader context of the pharmaceutical market and healthcare economics to facilitate a proper understanding of the implications of rising drug prices. It is crucial to consider the ownership of medications like oxycodone, which has been associated with the Sackler family/Purdue Pharma [16]. The ownership history raises significant questions regarding pharmaceutical industry practices, patent production, and the influence of monopolies. Additionally, heightened concerns regarding the addictive properties of oxycodone have led to stricter regulations and influenced public perception, potentially contributing to limited availability and subsequent price increases. The implications of such ownership emphasize the need to explore the broader societal and economic factors influencing drug pricing, encompassing concerns related to addiction and associated controversies. 

Although the number of beneficiaries for spinal fusion procedures has decreased by 17.7%, spinal fusion surgery costs have increased by 0.4%. This could partially be attributed to opioid-related complications such as an increased length of stay, respiratory depression, ileus, urinary retention, nausea, and vomiting [17]. Many patients who are prescribed opioids pre-operatively often require higher postoperative doses for longer periods [18]. Moreover, these opioid medication requirements can severely increase the risk of opioid-related adverse effects. Kalakoti et al. conveyed, in a cohort of 26,553 patients, that the strongest predictor for prolonged postoperative opioid use was preoperative use, with over half continuing to use at 1 year postoperatively [19]. This suggests that many of the patients are not only utilizing these medications immediately postoperatively but are continuing to use them long after their surgery is complete. This prolonged consumption can be another contributing factor to the rise in drug costs, as exemplified by Faour et al., who demonstrated a positive correlation between increased duration of opioid use and net medical cost per claim in spinal fusion surgeries [20].

Our analysis demonstrated that there was a significant negative correlation between the annual percent change in the total number of spinal procedure beneficiaries and the annual percent change in oxycodone costs, as well as a statistically significant negative correlation between the total number of spinal procedure beneficiaries and the cost of methocarbamol. A potential method to combat the rising costs of these drugs is the integration of a multimodal analgesia approach to pain management. These include preoperative, intraoperative, and postoperative interventions such as gabapentinoids, NSAIDS, local anesthetics, and neuromodulator medications, which have been found to improve pain control, reduce hospital narcotic consumption, and provide better functional outcomes [21,22,23]. The enhanced recovery after surgery program (ERAS) contains multimodal analgesics as part of the protocol to improve patient outcomes [23]. Additionally, ERAS protocols include non-pharmacological management of early postoperative mobilization and physical therapy as well as a prophylactic antiemetic pharmacological regimen aimed at preventing undesirable side effects. NSAIDs and COX-2 inhibitors, in the context of multimodal analgesics, can decrease the perioperative opioid prescription dose, cost, and length of stay in spinal fusion surgeries [24]. In the recent decade, the press towards generic drug availability has allowed for sounder fiscal alternatives, decreasing cost sharing for beneficiaries [25]. This decrease has occurred by way of increased competition, thereby driving prices down [26]. Lastly, shifting the surgical approach to minimally invasive techniques, such as transforaminal and percutaneous approaches for lumbar fusion surgery, has decreased inpatient hospital narcotic usage overall, including that of oxycodone [27]. Large-scale implementation of these techniques and guidelines may help alleviate rising drug costs driven by overprescription. 

In brain tumor resection surgeries, the number of beneficiaries decreased by 8.6%, and from the Pearson correlation analysis, there is a significant increase in percent change in cost and services for levetiracetam. It is known that the majority of neurosurgeons prescribe antiseizure medication prophylaxis for patients undergoing brain tumor resections; however, there are no clear clinical guidelines for correct use [28], which may cause heterogeneity in dosages based on provider preference, thus contributing to unneeded increased usage. Despite all of these findings, the use of antiseizure medications, such as levetiracetam, in brain tumor resections is still a widespread practice. Furthermore, the decreased number of beneficiaries may be linked to several factors, including the widespread use of antiseizure medications and the potential impact of medication shortages. In recent years, antiseizure medications have seen severe shortages, which may be attributed to the high market concentration of manufacturers and their limited capacity for spare production [29]. For example, 67.6% of patients on levetiracetam switched to different brands during these shortages, which may have ultimately led to price hikes. Another reason why the percent change in the cost of levetiracetam may have increased is due to its preferred use for its low side effect profile, unique mechanism, and low drug interactions compared to other antiseizure medications available [30]. Establishment of evidence-based guidelines for optimal dosing is needed, with potential avenues including extended-release dosing versus twice-a-day dosing [30]. Such guidelines could help standardize dosing practices and potentially reduce unnecessary usage, thereby mitigating the associated costs.

The advent of advanced pre-operative and intraoperative planning and imaging techniques has allowed for precise resection of brain tumors with smaller margins. Achieving gross total resection of brain tumors has been established as a favorable predictor of freedom from seizures postoperatively [31]. Optimizing tumor resection margins is a critical goal when performing these surgeries. Using multiple advanced imaging techniques intraoperatively, such as neuronavigation, Raman spectroscopy, and optical fluorescence, enhances the surgeon’s ability to achieve precise margins and accomplish the objective of total tumor resection [32]. Furthermore, utilizing these techniques decreases postoperative complications and allows for better prognostic outcomes in patients. Consequently, the implementation of these techniques has the potential to reduce the need for antiseizure medications by decreasing the likelihood of postoperative seizures, thus mitigating the requirement for prophylactic medication and potentially reducing associated costs.

Although we sought to comprehensively evaluate these trends, this study is not without limitations. First, the use of the Medicare Prescriber Public Use File Part D from the Centers for Medicare and Medicaid Services (CMS) from 2016 to 2020 as a database limited our results to patients who were mainly 65 years of age and older. As previously mentioned, although the greatest increase in elective lumbar fusion surgeries was in the elderly population of 65 and older, the mean age from 2004 to 2015 for lumbar fusion surgery was 59.0, with greater than 60% of the population coming from ages ranging from 20 to 65 [5]. Another limitation is the lack of variability in the procedures analyzed; it is possible that our analysis was not able to examine a wide range of prescriptions. There may be other prescriptions that are more commonly prescribed and are paid for by insurance companies that we were not able to include in our results since they are only focused on prescriptions paid for by Medicare. Lastly, another limitation of the database is that the four drugs mentioned have multiple indications that can be prescribed for various diseases. In the future, researchers can enhance their analysis of trends in drug usage for these surgeries by incorporating data from sources beyond Medicare. These additional sources could include private insurance databases or population-based surveys, enabling a more comprehensive and representative examination.

Additionally, Medicare and Medicaid use HCPCS billing codes. This could be a limitation in analyzing brain resection procedures because the codes may not accurately capture the full range of procedures that are performed in practice. Data analysis was performed using a linear regression model, which leaves our data sensitive to outliers and prone to underfitting. Finally, although our data suggest trends in cost associated with neurosurgical procedures, they do not take into account confounding factors such as the severity of the patient’s condition or variability in hospital-specific guidelines for drug usage in these procedures. These factors could significantly influence both the use of drugs and the overall costs involved. 

Lastly, our study failed to control potential confounding variables that could impact drug pricing. The pricing of prescription drugs in the U.S. is known to be unregulated compared to its peers, allowing companies to increase prices independently of demand. Additionally, the introduction of generic alternatives is known to influence drug prices and could act as a confounder in our analysis. In future studies, it is crucial to incorporate a comprehensive assessment by obtaining a more accurate understanding of the drivers of drug prices. By controlling for these, we can better isolate the effects of the variables of interest and draw more robust conclusions.

## 5. Conclusions

This study sought to examine the temporal and economic trends of commonly used drugs for both spinal fusion and brain tumor resection surgeries. Although there were multiple variables indicated to have been significantly correlated by Pearson’s correlation analysis, linear regression demonstrated that oxycodone was the only medication that had a significant association. Although oxycodone continues to play a huge role in spinal fusion procedures, the advent of multimodal anesthetic techniques, minimally invasive approaches to spinal fusion, and ERAS programs may reduce the use of opioids and, therefore, the cost of procedures. Additionally, implementing standardized clinical guidelines for levetiracetam dosing in brain tumor resection surgeries is critical to reducing its overuse. Advanced intraoperative imaging techniques allow surgeons to be more precise, leading to a better prognosis of seizure freedom while decreasing medication prophylaxis. As there are inherent limitations to this study, such as only examining Medicare Data, future studies should examine these factors using primary data from hospitals. This will lay the groundwork for reducing the overprescription of opioids, optimizing drug costs, and curbing the inflation of drug costs.

## Figures and Tables

**Figure 1 biomedicines-11-02185-f001:**
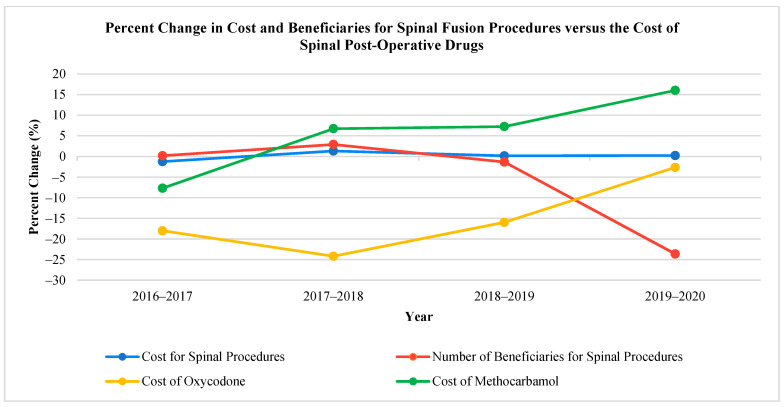
Percent difference of the total number of spinal fusion beneficiaries, the average cost of spinal fusion, and the cost of spinal fusion post-operative drugs.

**Figure 2 biomedicines-11-02185-f002:**
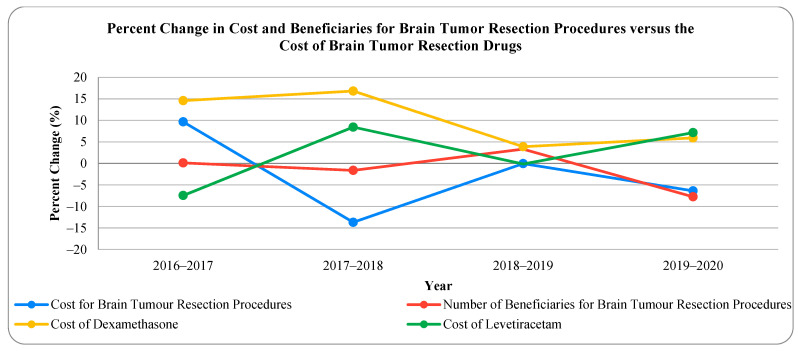
Percent difference in the total number of brain tumor resection beneficiaries, the average cost of brain tumor resection, and the cost of brain tumor resection drugs.

**Table 1 biomedicines-11-02185-t001:** The total number of prescriptions and average costs for the four drugs analyzed.

	Spinal Fusion Drugs				Brain Tumor Resection Drugs			
	Oxycodone		Methocarbamol		Dexamethasone		Levetiracetam	
Year	Number of Prescriptions	Average Cost	Number of Prescriptions	Average Cost	Number of Prescriptions	Average Cost	Number of Prescriptions	Average Cost
2020	68,171	$1471.23	14,042	$385.27	4837	$446.83	10,295	$1154.25
2019	75,757	$1510.29	11,044	$323.55	5024	$420.37	13,391	$1071.85
2018	87,978	$1751.57	7807	$300.12	5502	$404.08	12,754	$1073.46
2017	101,349	$2175.48	11,295	$279.90	5349	$336.13	12,329	$982.89
2016	107,663	$2567.31	11,037	$301.47	5292	$287.15	14,251	$1056.25

**Table 2 biomedicines-11-02185-t002:** The number of beneficiaries and the average cost for spinal fusion and brain tumor resection procedures by year.

	Spinal Fusion Procedures		Brain Tumor Resection	
Year	Beneficiaries	Average Submitted Cost	Beneficiaries	Average Submitted Cost
2020	65,987	$4937.06	2416	$9558.81
2019	81,591	$4926.18	2570	$10,301.00
2018	82,691	$4917.98	2571	$9959.76
2017	80,292	$4853.00	2923	$10,122.50
2016	80,156	$4914.08	2641	$10,111.51

**Table 3 biomedicines-11-02185-t003:** Regression coefficients for oxycodone.

Variable	B	95% CI	β	t	*p*
Total number of beneficiaries	0.180	[0.113, 0.248]	2.646	33.849	0.019 *
Total number of services	−0.134	[−0.184, −0.085]	−2.468	−34.290	0.019 *
Average Medicare payment	−3.059	[−17.301, 11.182]	−0.088	−2.730	0.224

R^2^adj = 0.997 (N = 5, *p* = 0.032). * indicates a significant finding. CI = confidence interval for B.

## Data Availability

Not applicable.

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
