# Peer review of "Economic Trends in Commonly Used Drugs for Spinal Fusion and Brain Tumor Resection: An Analysis of the Medicare Part D Database"

_biomedicines, 2023, doi:10.3390/biomedicines11082185_

Round 1
Reviewer 1 Report
The paper makes a significant attempt to understand the temporal and economic trends of drugs commonly used for spinal fusion and brain tumor resection surgeries. However, there are several limitations and weaknesses that hinder its credibility and applicability.
· The study only uses Medicare data, which limits the population scope. It excludes patients under 65 or those not covered by Medicare, which may lead to skewed results. Medicare patients typically have different health profiles, demographics, and socio-economic status than the rest of the population. Future studies should consider diversified data sources to present a more representative analysis.
· The study fails to control for potential confounders that could affect drug prices such as regulatory changes, market competition, the introduction of generic alternatives, and inflation rate. Not considering these factors may have resulted in an oversimplified model that fails to capture the complexity of drug pricing dynamics.
· The study overemphasizes the correlation between the number of beneficiaries and the cost of drugs, implying a causal relationship that may not exist. It is a basic tenet of research that correlation does not imply causation. Without robust causal inference methods, the conclusions drawn could be misleading.
· The reliance on linear regression may not be appropriate for this kind of analysis, especially if the relationships are not linear or if there are interaction effects between variables. Other statistical techniques, such as non-linear regression or machine learning algorithms, could potentially provide more accurate models.
· The choice of only two drugs for each procedure seems arbitrary and may limit the generalizability of the study. It would be more informative to look at a broader range of medications and consider their use in other types of surgeries as well.
· The study fails to situate its findings within the broader context of the pharmaceutical market or healthcare economics. This prevents a holistic understanding of the causes and implications of rising drug prices.
· While the paper does provide recommendations, it doesn't analyze their feasibility or potential impact. Suggestions like the implementation of standardized clinical guidelines need to be backed up by evidence of their efficacy, as well as a discussion of potential barriers to implementation.
Overall, while the paper takes an interesting approach to an important issue, it has critical flaws in its methodology and analysis. It offers a limited perspective and over-simplified interpretation of a complex problem, significantly undermining its potential contribution to the field. For future studies, it is recommended to adopt a more robust and comprehensive approach, with diversified data sources, more control variables, and a larger sample of drugs to offer a more representative and reliable analysis.
Reviewer 2 Report
This research correlated the use and price of Oxycodone, Methocarbamol, Dexamethasone, and Levetiracetam with spinal fusion and brain tumor resection surgeries. The manuscript is well written, and it is easy to read and to understand.
(1) The price of prescription drugs in USA is relatively unregulated as pharmaceutical companies can increase drug prices regardless of demand and independently of inflation rates. When pricing the drug several factors are taken into consideration, including the drug’s uniqueness, effectiveness, competition with other compounds, patients’ perception of value, and R&D (among others).
In the results sections (lines 164 to 209) several “r” values are shown. Could you please confirm that these are the R-squared coefficients of determination? Since the R-square is the percentage of the dependent (predicted) variable that a linear model explains, and the values that are shown are very high, I wonder about other factors that influence the price of a drug.
(2) Line 48. Could you please specify what subtypes of brain tumors haves increased in the recent years?
(3) Line 69. Regarding Alteplase. Could you please add “biosynthetic form of human tissue-type plasminogen activator (t-PA) with thrombolytic function” (or similar).
(4) Line 77. Regarding Oxycodone. Could you please add that it is a semi-synthetic opioid?
(5) Line 95. Could you please add the webpage? I think it may be “https://data.cms.gov/provider-summary-by-type-of-service/medicare-part-d-prescribers”.
(6) Lines 105 and 112. As I understand, in the data analysis the number of prescriptions for the 4 drugs (Oxycodone, Methocarbamol, Dexamethasone, and Levetiracetam) were obtained from the database, and the number of surgical procedures (spine fusion and brain tumor resections) as well. Nevertheless, these 4 drugs can also be used/prescribed for other diseases. Is it right?
(7) Line 119. Could you please specify what is the “another coefficient”?
(8) In Figures 1 and 2 the percent change is shown for surgical procedures and cost of drugs. Nevertheless, could you please explain how it was calculated? Which is the reference value? What is the formula? E.g. =(B2-A2)/A2?
(9) Regarding the data of 165 – 199. Is it possible to summarize the correlations using scatter plots with the fit line? Possibly both Pearson’s and Spearman’s correlations could be calculated.
(10) Could you please explain how the multiple linear regression analysis was setup? What are the predictors and the dependent (predicted) variables? Was the regression analysis calculated for each or all years together? What was the statistical test? Did you use SPSS, matlab, R, excel, etc.?
(11) Could you please upload the matrix with the data so other researchers can validate the statistical analyses? Is this possible/feasible/reasonable?
Author Response
Please see the attachment and Excel files.

Round 2
Reviewer 1 Report
The authors have addressed most of my comments well. However, there is still problem with statistical analysis. The statistical analysis in this research article, while thorough, does raise a few potential issues that should be addressed:
· The authors conducted a large number of Pearson correlation analyses between many variables. This increases the likelihood of obtaining statistically significant results purely by chance, an issue known as the multiple comparisons problem. Even with a strict alpha level (e.g., 0.05), if you conduct many tests, the probability of obtaining at least one significant result purely by chance increases. To address this, the authors should apply a correction method, such as the Bonferroni correction, to adjust the significance level.
· The authors seem to interpret the correlation as a form of causal relationship, which is a common misconception in research. The phrase "significantly predicted" may lead readers to believe that there is a causal relationship. However, correlation simply implies an association between variables, not causation. The authors need to clearly address this in their discussion to avoid misinterpretation of their findings.
· In the multiple linear regression analysis, the authors considered only the total number of beneficiaries, total number of services, and average Medicare payment as predictors for drug cost. However, drug costs can be influenced by a multitude of factors such as production costs, market demand, and pricing policies. The lack of these control variables in the regression model could lead to omitted variable bias and inaccurate results.
· The authors used Pearson correlation which assumes that the variables are normally distributed and that the relationship between the variables is linear. However, the authors did not report checking these assumptions. If the assumptions are not met, the results may be inaccurate.
· Despite reporting a number of significant findings, the authors did not mention the power of their study or conduct a post-hoc power analysis. Given the large number of tests conducted, a power analysis would be helpful to ensure the study is adequately powered to detect true significant effects.
· The authors do not report any effect sizes for their findings. Reporting only the significance of the findings (p-values) does not provide information on how large or important the effect is. Therefore, it would be beneficial to include effect sizes along with their confidence intervals to give a complete picture of the findings.
In conclusion, the authors need to consider the issues raised and reanalyze their data taking into account these concerns to ensure the validity and reliability of their findings.
Round 3
Reviewer 1 Report
The authors have addressed all issues and problems raised in my review. I recommend to accept.